# Drivers of the opioid crisis: An appraisal of financial conflicts of interest in clinical practice guideline panels at the peak of opioid prescribing

Sheryl Spithoff[1,2]*, Pamela Leece[2,3,4], Frank Sullivan[2,5], Nav Persaud[2,6], Peter Belesiotis[7], Liane Steiner[8]

**1** Department of Family and Community Medicine, Women's College Hospital, Toronto, Canada, **2** Department of Family and Community Medicine, University of Toronto, Toronto, Canada, **3** Dalla Lana School of Public Health, University of Toronto, Toronto, Canada, **4** Public Health Ontario, Toronto, Canada, **5** Medical School University of St Andrews, St Andrews, Scotland, United Kingdom, **6** Department of Family and Community Medicine, St. Michael's Hospital, Toronto, Canada, **7** Faculty of Health Sciences, McMaster University, Hamilton, Canada, **8** St. Michael's Hospital Centre for Urban Health Solutions, Toronto, Canada

\* sheryl.spithoff@wchospital.ca

**Data Availability Statement:** All relevant data are available from http://hdl.handle.net/1807/98896.

**Funding:** The authors received no specific funding for this work.

## Abstract

### Background

Starting in the late 1990s, the pharmaceutical industry sought to increase prescribing of opioids for chronic non-cancer pain. Influencing the content of clinical practice guidelines may have been one strategy industry employed. In this study we assessed potential risk of bias from financial conflicts of interest with the pharmaceutical industry in guidelines for opioid prescribing for chronic non-cancer pain published between 2007 and 2013, the peak of opioid prescribing.

### Methods

We used the Guideline Panel Review (GPR) to appraise the guidelines included in the 2014 systematic review and critical appraisal by Nuckols *et al*. These were English language opioid prescribing guidelines for adults with chronic non-cancer pain published between July 2007 and July 2013, the peak of opioid prescribing. The GPR assigns red flags to items known to introduce potential bias from financial conflicts of interest. We operationalized the GPR by creating specific definitions for each red flag. Two reviewers independently evaluated each guideline. Disagreements were resolved with discussion. We also compared our score to the critical appraisal scores for overall quality from the study by Nuckols *et al*.

### Results

We appraised 13 guidelines, which received 43 red flags in total. Guidelines had 3.3 red flags on average (out of a possible seven) with range from one to six. Four guidelines had missing information, so red flags may be higher than reported. The guidelines with the

**Competing interests:** The authors have declared that no competing interests exist.

highest and second highest scores for overall quality in the 2014 critical appraisal by Nuckols *et al.* had five and three red flags, respectively.

## Conclusion

Our findings reveal that the guidelines for opioid prescribing chronic non-cancer pain from 2007 to 2013 were at risk of bias because of pervasive conflicts of interest with the pharmaceutical industry and a paucity of mechanisms to address bias. Even highly-rated guidelines examined in a 2014 systematic review and critical appraisal had many red flags.

## Introduction

Understanding the root causes of the opioid crisis may help prevent similar iatrogenic epidemics in the future. The pharmaceutical industry's influence on physician prescribing, particularly in Canada and the US, appears to be a major cause of the crisis [1,2]. In the late 1990s, Purdue Pharma started aggressively marketing its opioid, Oxycontin (oxycodone), for chronic non-cancer pain (CNCP) through academic detailing and education sessions for physicians [3]. Other pharmaceutical companies followed suit [4]. These activities have been linked to the rise in opioid prescriptions and the subsequent harms [3–7].

Clinical practice guidelines are another mechanism that the pharmaceutical industry may have used to influence physicians' opioid prescribing practices. Clinical practice guidelines provide evidence-based clinical recommendations to improve patient care and outcomes [8–10]. They have a moderate impact on the behaviour of health care providers [11–13]. A recent study found that the 2016 Centre for Disease Control (CDC) guideline for opioid prescribing led to declines in opioid prescribing [14]. Given the potential impact of guidelines, the Institute of Medicine in its 2011 report "Clinical Practice Guidelines We Can Trust," recommends that organizations that produce guidelines take steps to mitigate bias. These steps include selecting guideline chairs and committee members without financial conflicts of interest, creating a multi-disciplinary committee and ensuring a rigorous external review [8]. These recommendations aligned with earlier guidance [15]. Research has demonstrated, however, that many guidelines do not adhere to these recommendations, particularly independence from the pharmaceutical industry [16–19].

To date, guidelines for opioid prescribing for CNCP have escaped scrutiny. For example, in 2014, well into the opioid crisis and at the peak of opioid prescribing [20–24], Nuckols *et al.* published a systematic review and critical appraisal of guidelines for opioid prescribing for adults with CNCP. Other than reporting the range of scores for editorial independence, it did not address potential bias from financial conflicts of interest with the pharmaceutical industry [25].

Our study objective was to assess the potential risk of bias from financial conflicts of interest with the pharmaceutical industry in the guidelines for opioid prescribing for chronic non-cancer pain that were included in the 2014 systematic review and critical appraisal by Nuckols *et al.* [25].

## Methods

### Overview

We conducted an appraisal of the 13 clinical practice guidelines included in the 2014 systematic review and critical appraisal by Nuckols *et al.* [25]. (Table 1). For each guideline we

**Table 1. Location, sponsor and name of the 13 guidelines for opioid prescribing for chronic non-cancer pain (CNCP) included in Nuckols *et al.* 2014 systematic review and critical appraisal [25].**

| Country | Sponsor/Authors | Name of guideline | Reference |
|---------|-----------------|-------------------|-----------|
| U.S. | American College of Occupational and Environmental Medicine (ACOEM) | ACOEM Guidelines for Chronic Use of Opioids (2011) | [26] |
| U.S. | American Geriatrics Society (AGS) | Pharmacological Management of Persistent Pain in Older Persons and Management of Persistent Pain in Older Persons (2009) | [27,28] |
| U.S. | American Pain Society and American Academy of Pain Medicine (APS and AAPM) | Clinical Guidelines for the Use of Chronic Opioid Therapy in Chronic Noncancer Pain (2009) | [29–31] |
| U.S. | American Society of Anesthesiologists (ASA) | Practice Guidelines for Chronic Pain Management: An Updated Report by the American Society of Anesthesiologists Task Force on Chronic Pain Management and the American Society of Regional Anesthesia and Pain Medicine (2010) | [32] |
| Canada | American Society of Interventional Pain Physicians (ASIPP) | American Society of Interventional Pain Physicians (ASIPP) Guidelines for Responsible Opioid Prescribing in Chronic Non-Cancer Pain (2012) | [33,34] |
| U.S. | National Opioid Use Guideline Group (NOUGG) | Canadian Guideline for Safe and Effective Use of Opioids for Chronic Non-Cancer Pain (2010) | [35–38] |
| U.S. | Colorado Division of Workers' Compensation (Colorado DWC) | Chronic Pain Disorder Medical Treatment Guidelines (2011) | [39] |
| U.S. | Fine et al, 2009 | Establishing "Best Practices" for Opioid Rotation: Conclusions of an Expert Panel (2009) | [40] |
| U.S. | Institute for Clinical Systems Improvement (ICSI) | Assessment and Management of Chronic Pain (2011) | [41] |
| U.S. | University of Michigan Health System (UMHS) | Managing Chronic Non-Terminal Pain in Adults, Including Prescribing Controlled Substances (2012) | [42] |
| U.S. | Utah Department of Health (UDOH) | Utah Clinical Guidelines on Prescribing Opioids for Treatment of Pain (2009) | [43,44] |
| U.S. | Veterans Affairs and Department of Defense (VA/DoD) | Clinical Practice Guideline for Management of Opioid Therapy for Chronic Pain (2010) | [45] |
| U.S. | Work Loss Data Institute (WLDI) | Pain (Chronic) (2011) | [46] |

compared our findings to their critical appraisal score for overall guideline quality. This approach allowed us to build upon Nuckols *et al.*'s work.

## Data sources

We assessed the 13 guidelines for opioid prescribing for CNCP that were included in the Nuckols *et al.* 2014 systematic review and critical appraisal [25]. Nuckols *et al.* included English language guidelines published between 2007 and 2013 that addressed the use of opioids to treat CNCP in adults. The guidelines had to be clinical practice guidelines that included "*recommendations intended to optimize patient care that are informed by a systematic review of evidence and an assessment of the benefits and harms of alternative care options*" and were published after 2006 [47]. To assess the quality of the Nuckols review, two authors (SMS and PL) independently rated the guidelines using the JBI Critical Appraisal Checklist for Systematic Reviews and Research Syntheses [48]. Differences were resolved with discussion. The Nuckols review satisfied all checklist criteria except for item 7: "*Were there methods to minimize errors in data extraction?*" For item 7 the authors do not mention any methods to minimize errors in data extraction (e.g. data extraction in duplication or extraction double-checked). There were no concerns about the inclusion criteria, search strategy, criteria for appraising studies, appraisal method and methods to combine the data.

For each guideline, we evaluated the same version that was reviewed and appraisal by Nuckols *et al.* In two cases we had to request the version of the guideline used by Nuckols *et al.* from the sponsoring organization because it was no longer available online [41,42]. If the guideline did not contain information on the chair and committee members' conflicts of interest we contacted the sponsoring organization or authors for this information. We also used supplementary information on guideline methods when available from the sponsoring organization

[49,50]. Since individuals and organizations often neglect to report conflicts of interest [51–53], we checked independent sources. To determine if the sponsoring organization was a professional society that receives pharmaceutical industry funding or was a proprietary company, two reviewers independently searched the organization's website (including any webpages on sponsorship and annual reports) to determine if it received funds from the pharmaceutical industry. To determine if the committee chair had conflicts of interest in the two years prior to publication, two reviewers independently conducted a Pubmed search and a Google search of the chair's name to look for this information. For the Pubmed search we searched the author's name for any publications for the same year the guideline was published and checked if these listed any conflicts of interest. For the Google search, we searched the author's name with the term "conflict of interest." We reviewed the first 30 entries (three pages) for information on conflicts at time of guideline publication and up to two years prior. We opted for conflicts of interest up to two year prior because that was a common standard for conflict of interest reporting in 2007 to 2013 [15].

For their critical appraisal Nuckols *et al.* assessed guideline quality using the Appraisal of Guidelines for Research and Evaluation II (AGREE II) instrument, [54] and the systematic review supporting each guideline [25] using the A Measurement Tool to Assess Systematic Reviews (AMSTAR) tool [55]. The AGREE II instrument assesses the quality and reporting of practice guidelines [54], and it can also be used to inform guideline development and reporting [56].

## Instrument

To guide our assessment of the risk of bias in guidelines, we used the Guideline Panel Review (GPR) [57]. The GPR was created by an international expert working group (the Guideline Panel Review working group) in 2013 using a modified Delphi process [57]. The tool assigns red flags to practice elements "*known to introduce potential bias*" [57]. The goal of the group was to inform patients and those that develop, publish and use guidelines. The GPR reflects the standards created by the Institute of Medicine for developing trustworthy guidelines [8]. The GPR focuses on financial conflicts of interest, in particular those involving industry, because industry-funded networks dramatically amplify findings that support its products and obscure those that do not [57,58].

## Operationalizing the GPR

The GPR has not been validated or operationalized and at present is described by the creators as a "framework for future developments" [57]. The developers also designed the tool to be used by the organization producing the guidelines and submitted to the journal along with the guidelines. Journals would use the information to decide whether or not to publish the guidelines. They would also publish the GPR along with the guideline. We found, however, that it was able to also provide us with a framework as we appraised the guidelines. Lenzer and colleagues also used this approach to evaluate guidelines [57]. We operationalized the GPR for our purposes by creating specific definitions for each red flag (Table 2) through several rounds of testing and discussion. Differences were resolved through discussion. Our operationalized statements vary in several places from how the GPR was employed by Lenzer and colleagues [57]. For financial conflicts of interest, we report only on financial conflicts of interest with industry. Other financial conflicts of interest can have an impact—like professional conflicts—however, this was not the focus of our study. The most challenging red flag to operationalize was panel or committee stacking because it was very difficult to determine each committee member's views prior to the guideline development. This may be easily accomplished if

**Table 2. Description of items on the Guideline Panel Review (GPR) tool and GPR elements known to introduce bias [57], and our operationalized and modified statement.**

| Item on the GPR | Element known to introduce potential bias | Operationalized and modified statement |
|---|---|---|
| Sponsor | Sponsor(s) is a professional society that receives substantial industry funding or sponsor is a proprietary company, or is undeclared or hidden | Sponsor is a professional society that receives pharmaceutical industry funding or is a proprietary company |
| Committee chair (s) | Committee chair(s) have any financial conflict | Financial conflicts of interest with the pharmaceutical industry at time of publication or within two years prior |
| Committee members | Multiple panel members have any financial conflict | Multiple committee members with financial conflicts of interest with the pharmaceutical industry |
| Committee stacking | Any suggestion of committee stacking that would pre-ordain a recommendation regarding a controversial topic | Less than 10% committee members with pain or addiction expertise |
| Role of methodologist | No or limited involvement of an expert in methodology in the evaluation of evidence | No methodologist or methodologist has a minor role (i.e., not the chair or lead of the committee or lack of an oversight methodology committee) |
| External review | No external review | No formal external review by unaffiliated individuals or groups |
| Committee composition | No inclusion of non-physician experts/patient representative/ community stakeholders | No non-physicians OR no patient representatives (must have both) |

guideline sponsors asked participants to self-report their views. However, it was not feasible for our assessment. Therefore, in discussion, we decided we would use expertise in chronic pain or addictions as a proxy for a balanced committee. This would ensure there were committee members who saw the benefits of opioid prescribing for CNCP and members who saw the harms. We set a minimum of 10% of committee members from each of these two categories. And finally, the GPR also uses a "caution" rating for items that may be an "*important part of guideline development, but for which there is not proof that bias is introduced by the presence of that element.*" We opted not to report cautions because of the lack of evidence to support these items.

## Data extraction

We created a data abstraction form using the GPR. All six reviewers independently pilot-tested the form on two guidelines. We reviewed the results and modified the form based on team feedback. Two reviewers then independently assessed each guideline and completed the data abstraction form. Differences in opinion were resolved through discussions, and when needed, a third reviewer (SMS) assisted in making the final decision.

## Outputs

Our outputs included: name of the guideline and sponsoring organization; location; date published; GPR red flags; sources for completion of GPR (official guideline/supporting documents/information on website/correspondence with the sponsoring organization and/or authors producing the guideline). We compared the AGREE II scores for overall quality from Nuckols *et al.'s* systematic review and critical appraisal [25] to our findings. The overall quality score is a global score reported by reviewers after rating all the domains of quality. It is based on the other scores but is not calculated from those scores [25,54]. We did not compare our findings to the AMSTAR scores from the study because AMSTAR assesses systematic reviews underpinning the guidelines, not the guideline itself.

## Data synthesis

We displayed our results in tabular format to summarize the number of red flags and uncertain items (items which could not be confidently appraised with available information) for

each item. For each guideline we reported on the total number of red flags and the AGREE II scores for overall quality from Nuckols *et al.'s* review [25].

## Results

### Overview

We appraised the 13 guidelines in the systematic review and critical appraisal by Nuckols *et al.* (Table 1). The guidelines, one Canadian and the rest American, were published between 2009 and 2012. We assigned 43 red flags in total to the 13 guidelines (Table 3).

### Sponsor

We assigned a red flag to four of the guidelines because the sponsoring organization was a professional society that accepted funds from the pharmaceutical industry [27,31,32,45], and a red flag to the fifth one because the guideline was funded through an unrestricted pharmaceutical grant [40]. Among the four professional societies that accepted pharmaceutical industry funding, this information was not declared in the guidelines, but was found in supplementary sources.

### Chair(s)

We assigned a red flag to five guidelines because at least one of the committee chairs received funding from the pharmaceutical industry in the two years prior to its publication [27,30, 35,40], all but one [35] from opioid manufacturers. We also assigned a red flag to a guideline because the co-chair was on the board of directors of patient advocacy group that received over 90% of its funding from the pharmaceutical industry [45]. In three of the five guidelines that received a red flag, conflicts of interest information was either not declared, or not stated in the guideline or available from the sponsoring organization, but were found in other source documents [27,40,45]. We assigned an uncertain rating to one guideline because its sponsor did not report the names of committee chairs, and we were unable to contact the organization [39].

**Table 3. Guideline Panel Review (GPR) [57] red flags (indicating potential bias) in the opioid prescribing for chronic non-cancer pain (CNCP) guidelines included in Nuckols *et al.* 2014 systematic review and critical appraisal [25].**

| Item on the GPR | Element known to introduce potential bias | # of guidelines with a red flag | # of guidelines with uncertain* items |
|---|---|---|---|
| Sponsor | Sponsor is a professional society that receives pharmaceutical industry funding or is a proprietary company | 5 | 0 |
| Committee chair(s) | Financial conflicts of interest with the pharmaceutical industry at time of publication or within two years prior | 5 | 1 |
| Committee members | Multiple committee members with financial conflicts of interest with the pharmaceutical industry | 3 | 4 |
| Committee stacking | Less than 10% committee members with pain or addiction expertise | 7 | 1 |
| Role of methodologist | No methodologist or methodologist has a minor role (i.e., not the chair or lead of the committee or an oversight committee) | 7 | 0 |
| External review | No formal external review by unaffiliated individuals or groups | 4 | 0 |
| Committee composition | No non-physicians OR no patient representatives (must have both) | 12 | 0 |
| Total | | 43 | 6 |

* Element that could not be confidently appraised with available information

### Committee members

We assigned a red flag to three guidelines because at least two of the committee members had received funding from the pharmaceutical industry in the two years prior to its publication [27,29,35]. We assigned an uncertain rating to four guidelines because they did not contain conflict of interest declarations, and we were either unable to contact the sponsor or authors [39,40]; or the sponsoring organization declined to provide the information [32]; or only provided limited information [45].

### Panel stacking

We assigned a red flag to six guidelines because they had evidence of "panel stacking:" the exclusion of members likely to have an opposing view [26,29,32,41,42,46]. These six guidelines had less than 10% of their members with expertise in addictions. For one guideline we were not able to determine if there was panel stacking because information on the committee members was not included in the guideline and we were not able to contact the author or guideline sponsor [40].

### Methodologist

We assigned a red flag to seven guidelines because no methodologist was mentioned, or they were not a chair or part of an oversight committee for methodology [27,33,39–42,46].

### External review

We assigned a red flag to four guidelines because they did not undergo a formal external review [32,39,40,42]. For two guidelines, information on the external review was located in supplementary materials [26,45].

### Composition

Finally, only one guideline committee was both multi-disciplinary and included patient representatives, and therefore was not assigned a red flag [41].

### Individual guidelines

The number of red flags for an individual guideline ranged from one to six (out of a possible seven) (see Table 4 for more details). Mean and median red number of flags for guidelines were 3.3 and 3.0, respectively. Since we were unable to obtain conflict of interest information from the authors or sponsoring organization of four guidelines, the number of red flags are likely higher than reported. The guidelines with the four highest AGREE II scores for overall quality [30][36][26][45] had five, three, two and three red flags, respectively. The guidelines with the lowest four AGREEII scores for overall quality [32][39][40][46], had four, three, five and three red flags, respectively.

## Discussion

Our appraisal demonstrates that the pharmaceutical industry had a pervasive presence in clinical practice guidelines for opioid prescribing for CNCP from 2007 to 2013, the peak of opioid prescribing [20–24]. Additionally, organizations that produced the guidelines failed to regularly employ mechanisms, such as appointing a methodologist in a lead role or conducting an external review—to mitigate potential bias from industry involvement. Even guidelines that had high AGREE II scores for overall quality (indicating rigour in development and reporting

**Table 4. Guideline Panel Review (GPR) [57] red flags (indicating potential bias) in the opioid prescribing for chronic non-cancer pain (CNCP) guidelines included in Nuckols *et al*. 2014 systematic review and critical appraisal [25].**

| Guideline | Total # Red flags* (scale: 0–7) | Total # Uncertain items** (scale: 0–7) |
|---|---|---|
| American College of Occupational and Environmental Medicine (ACOEM), 2011 | 2 | 0 |
| American Geriatrics Society (AGS), 2009 | 5 | 0 |
| American Pain Society-American Academy of Pain Medicine (APS-AAPM), 2009 | 5 | 0 |
| American Society of Anesthesiologists (ASA), 2010 | 4 | 1 |
| American Society of Interventional Pain Physicians (ASIPP), 2012 | 3 | 0 |
| National Opioid Use Guideline Group (NOUGG), 2010 | 3 | 0 |
| Colorado Division of Workers' Compensation (DWC), 2011 | 3 | 2 |
| Fine et al, 2009 | 5 | 2 |
| Institute for Clinical Systems Improvement (ICSI), 2011 | 2 | 0 |
| University of Michigan Health System (UMHS), 2012 | 4 | 0 |
| Utah Department of Health (UDOH) 2009 | 1 | 0 |
| Veterans Affairs/Department of Defense (Va/DoD), 2010 | 3 | 1 |
| Work Loss Data Institute (WLDI), 2011 | 3 | 0 |

*Element known to introduce potential bias

**Element that could not be confidently appraised with available information

of the guideline) (20), had many red flags in our appraisal because of potential bias. Many guidelines also had missing or incomplete information on the sponsoring organization's funding sources, and on panel members' conflicts of interest. And finally, in a number of cases, guidelines provided incomplete or inaccurate information about financial conflicts of interest. As a result, the appraisal likely provides a conservative estimate of the potential for bias.

To our knowledge, this is the first study to demonstrate the pervasive presence of the pharmaceutical industry in guidelines for opioid prescribing for CNCP. This aligns with past research showing the pharmaceutical industry's widespread involvement in guidelines on other topics [16,17,19,59,60]. A 2011 systematic review of studies that examined the effect of conflicts of interest on guidelines development and recommendations, found that most guidelines had committee members with conflicts of interest [59]. These conflicts of interest with the pharmaceutical industry can affect the attitudes and behaviour of individuals and organizations. Individuals with more conflicts of interest with pharmaceutical companies are more likely to espouse industry-friendly positions and prescribe more drugs [61–63]. Organizations with more financial conflicts of interest are more likely to make recommendations that would benefit industry [57,64,65]. Therefore, bias in guidelines for opioid prescribing for CNCP may have led to industry-friendly recommendations contributed to high rates of opioid prescribing. These findings are particularly concerning because risk of bias from industry presence in opioid prescribing guidelines appears to be ongoing; the 2017 Canadian Guideline for Opioids for Chronic Non-Cancer Pain had several members with conflicts of interest, including one who had been a speaker for Purdue Pharma [66,67].

Our finding that guidelines that scored high for overall quality on the 2014 critical appraisal by Nuckols *et al*. [25] had many red flags aligns with findings in a study by Eady *et al*. [68]. In an appraisal of guidelines on treating acne, the authors reported that using the AGREE II tool

during guideline development did not have "as great an effect on guideline quality as expected. There is considerable room for improvement in acne treatment guidelines in order to satisfy the Institute of Medicine's trustworthiness criteria and avoid bias." The AGREE II tool assesses conflicts of interests and strategies to mitigate bias differently than the GPR and the Institute of Medicine. The GPR assigns a red flag if the sponsoring organization or individuals on the guideline committees have a financial conflict of interest. The AGREE II, however, rates a guideline with conflicts of interest equally as one without, as long as they are reported and their potential impact on the guideline described [54]. The medical literature indicates, however, that transparency is not sufficient to reduce bias and divestment from financial conflict is a better strategy [8,69–72]. Additionally, Nuckols *et al.* did not use supplementary sources for conflicts of interests and therefore did not identify the four sponsoring organizations and three chairs who did not accurately report financial conflicts of interest with the pharmaceutical industry. This may have affected the AGREE II score for overall quality. Finally, the AGREE II tool gives a guideline with a methodologist in a minor role a high score, whereas the GPR requires that the methodologist have a major role. The literature supports having a methodologist in a major role, because when content experts take the lead, evidence reviews are less accurate [73–75]. These findings indicate that the AGREE II may not accurately assess risk of bias from financial conflicts of interest with industry, particularly if appraisers do not seek out supplementary sources to assess for hidden conflicts of interest.

Our finding that many guidelines either did not report or accurately report funding sources for the sponsoring organization and conflicts of interest for individuals is consistent with other studies and media reports [53,76–80]. A recent ProPublica and New York Times investigation found that a top cancer researcher consistently failed to disclose millions in payments [81]. Although some of the guidelines in our study were published prior to the 2011 recommendations from the Institute of Medicine, accurate disclosure was the accepted standard during the time period of these guidelines were published [82]. Reasons that individuals and organizations do not declare conflicts of interest has been poorly studied, but may include: not understanding what to declare [83,84]; believing that a conflict is irrelevant [85]; and concerns about reducing trust in the guideline user [85]. Independent verification of conflicts of interest would help address reporting inaccuracies. This is easily done in the US with the Open Payments database (available since 2014) that records all industry payments to physicians [80]. A similar database, however, does not exist in Canada [86].

## Limitations

Our study allowed us to build on the past work of Nuckols *et al.* However, it also means we did not create and conduct our own search strategy. In particular, the English language requirement may have excluded some guidelines that would have otherwise met the inclusion criteria. It is also possible that Nuckols *et al.* missed English language guidelines (despite an appropriate search strategy). Additionally, although we used the same version of the guidelines as in the systematic review by Nuckols *et al.*, we did not access the documents on the same dates. In two cases we had to order the version of the guideline used by Nuckols *et al.* from the sponsoring organization because it was no longer available online [41,42]. Therefore, it is possible that the sources were slightly different and this may have affected our findings. The GPR also has some limitations. It is based on recommendations from the Institute of Medicine and other evidence [8,57], but has not been used frequently to date and is not a standardized or validated tool. However, it does provide a structured approach to descriptively appraise clinical practice guidelines. Another limitation was our difficulty in assessing for panel stacking. The GPR was designed for use by the guidelines organizer who would complete and submit the GPR along

with the guidelines for publication. In this situation, the committee could easily and accurately assess the important views of the committee members prior to the creation of the guidelines. Conducting the assessment as external reviewers is much more difficult. However, we believe our approach (ensuring an adequate number of chronic pain and addictions specialists) would ensure that there were people on both sides of the opioid prescribing debate. Another limitation is our Google search for conflicts of interest not reported in the guidelines. It is possible that we missed conflicts of interest by not conducting a more exhaustive search. Therefore, our study is likely a conservative estimate of the potential for bias. And a final limitation is that we only examined conflicts of interest with the pharmaceutical industry. We did not assess other financial conflicts of interest (e.g. an individual's income from performing surgical procedures) [87][57], which can lead to bias in guidelines, but these were not the focus of our study.

## Conclusion and next steps

Our findings reveal that the clinical practice guidelines for opioid prescribing for CNCP from 2007 to 2013 were at risk of bias because of pervasive conflicts of interest with the pharmaceutical industry, and with a paucity of mechanisms to mitigate bias. Even highly rated guidelines in a 2014 systematic review and critical appraisal had many red flags indicating a high risk of bias. More research is needed to understand the impact of the pharmaceutical industry's presence via conflicts of interest in opioid prescribing guidelines. This is particularly important given recent evidence of ongoing industry involvement in guidelines for opioid prescribing for CNCP.

Guideline sponsors, researchers and guideline users should consider using the GPR in guideline development and appraisal to assess risk of bias. Guideline developers should search independent sources where available, such as the Open Payments database, to verify the conflicts of interest reported by committee chairs and members.

## Supporting information

**S1 Table. Master Data Extraction Sheet.** Table with information on sponsor/author, guidelines name, references, elements known to introduce bias, number of red flags and uncertain rating and notes for each guideline.
(DOCX)

## Acknowledgments

We would like to thank Susan Hum, Research Associate (Women's College Hospital Department of Family and Community Medicine) for reviewing and editing this paper.

## Author Contributions

**Conceptualization:** Sheryl Spithoff, Pamela Leece, Frank Sullivan, Nav Persaud.

**Formal analysis:** Sheryl Spithoff, Pamela Leece, Frank Sullivan, Nav Persaud, Peter Belesiotis, Liane Steiner.

**Writing – original draft:** Sheryl Spithoff.

**Writing – review & editing:** Pamela Leece, Frank Sullivan, Nav Persaud, Peter Belesiotis, Liane Steiner.

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
