## [Decision Letter · Decision Letter 0]

21 Jul 2019

PONE-D-19-17372

Roots of the opioid crisis: an appraisal of financial conflicts of interest in clinical practice guideline panels at the peak of opioid prescribing

PLOS ONE

Dear Dr. Spithoff,

Thank you for submitting your manuscript to PLOS ONE. After careful consideration, we feel that it has merit but does not fully meet PLOS ONE’s publication criteria as it currently stands. Therefore, we invite you to submit a revised version of the manuscript that addresses the points raised during the review process.

In addition to the comments from the three reviewers I have some additional points that need to be addressed:

Do you have any information that these guidelines actually influenced prescribing behaviour? If they were largely ignored then it shouldn’t matter whether they were biased. On a related topic, if the guidelines influenced prescribing were they all equally influential? Both the GPR and the IOM recommendations came in after 2007. Were there recommendations about FCOI of panel members before that that the guidelines should have followed?How were the “red flag” definitions developed?

We would appreciate receiving your revised manuscript by Sep 04 2019 11:59PM. To enhance the reproducibility of your results, we recommend that if applicable you deposit your laboratory protocols in protocols.io, where a protocol can be assigned its own identifier (DOI) such that it can be cited independently in the future. For instructions see: http://journals.plos.org/plosone/s/submission-guidelines#loc-laboratory-protocols

We look forward to receiving your revised manuscript.

Kind regards,

Joel Lexchin, MD

Academic Editor

PLOS ONE

Journal Requirements:

1. We note that you have stated that you will provide repository information for your data at acceptance. Should your manuscript be accepted for publication, we will hold it until you provide the relevant accession numbers or DOIs necessary to access your data. If you wish to make changes to your Data Availability statement, please describe these changes in your cover letter and we will update your Data Availability statement to reflect the information you provide.

2. Please ensure that you refer to Figure 1 in your text as, if accepted, production will need this reference to link the reader to the figure.

3. We note that Figure(s) [1] in your submission contain copyrighted images. All PLOS content is published under the Creative Commons Attribution License (CC BY 4.0), which means that the manuscript, images, and Supporting Information files will be freely available online, and any third party is permitted to access, download, copy, distribute, and use these materials in any way, even commercially, with proper attribution. For more information, see our copyright guidelines: http://journals.plos.org/plosone/s/licenses-and-copyright.

1.    You may seek permission from the original copyright holder of Figure(s) [1] to publish the content specifically under the CC BY 4.0 license.

Reviewers' comments:

Reviewer's Responses to Questions

**Comments to the Author**

1. Is the manuscript technically sound, and do the data support the conclusions?

Reviewer #1: Partly

Reviewer #2: Yes

Reviewer #3: Partly

2. Has the statistical analysis been performed appropriately and rigorously? 

Reviewer #1: N/A

Reviewer #2: N/A

Reviewer #3: N/A

3. Have the authors made all data underlying the findings in their manuscript fully available?

Reviewer #1: No

Reviewer #2: Yes

Reviewer #3: Yes

4. Is the manuscript presented in an intelligible fashion and written in standard English?

Reviewer #1: No

Reviewer #2: Yes

Reviewer #3: Yes

5. Review Comments to the Author

Reviewer #1: This paper addresses a gap in the literature examining pharmaceutical industry involvement in the opioid epidemic – industry influence on clinical practice guidelines. It builds on a previous study examining opioid prescribing guidelines (for chronic non-cancer pain). As this earlier study had a number of weaknesses and the current study methods are vague, I think the paper needs substantial clarification before publication.

Introduction

Good description of the rationale for this paper. However, as this paper relies on the Nuckols review, I would like to see a more detailed critique of the pros and cons of this review. Why was it selected as the starting point, rather than the authors doing their own search and assessment of guidelines? Furthermore, this limits the study to only North American guidelines.. this needs to be justified.

Methods

It is difficult to evaluate the search strategy for guidelines as it is based on the Nuckols paper. I looked at the Nuckols paper and it is surprising they identified so few guidelines to include given their search strategy.

There needs to be much more description of the GPR as this would help explain some of the decisions made about how nondisclosed financial ties were identified. For example, page 6, line 124, it is not clear why searches for undisclosed conflicts of interests were made 2 years prior for the committee chairs and only in the current year for the committee members (line 127). Many studies look for undisclosed COI for 3 years prior to the index guideline (see Moynihan re undisclosed ties in NHMRC guidelines). The results (page 9) mention only the 2 year period.

Table 2 shows how the authors operationalized the “red flags” in the GPR, but these are not the same as criteria used for the GPR. There should be a description of how the categories used in this paper were derived.

Page 7, line 157 mentioned a GPR score, but no scores are calculated or reported; the number of red flags is reported.

Page 7, line 161. The authors should state why they extracted only the AGREEII scores and not the AMSTAR scores (presumably because AMSTAR applies to the systematic reviews used in the guidelines). In addition, I am not sure that it is allowable for the authors to reprint the AMSTAR scores from Nuckols in this paper without permission. Nuckols is not published in an open access journal.

Results

My concerns with the results are primarily related to the use of the GPR.

Table 2 needs clarification. Table 2 should include the N in the title, which I assume is 13 guidelines? So, if I am interpreting the table correctly, 5 of 13 guidelines had a “red flag” for sponsor, meaning that the sponsor was a professional society that receives pharmaceutical industry funding or is a proprietary company.

I disagree with the author’s definition of “Panel stacking” so think this needs further justification. I would argue that having panel members from multiple disciplines would promote diversity of views and minimize bias. However, the author rating suggests that there should be more members with expertise in addictions.

The authors should define a major vs a minor role for a methodologist, especially as this issue is raised again in comparison to the AGREEII tool (page 12, lines 290-294).

Conclusion

The comparison with the AGREEII tool, which highlights the AGREE limitations, is important and should be consolidated in one place in the discussion.

I would also highlight the extent of nondisclosure in the guidelines reviewed.

Reviewer #2: I have provided comments in the attachment. My only comment here would be pertaining to #1 - that the authors should be cautious in their wording about bias and influence - the paper does not measure bias or influence (i.e., the real effects of the FCOI relationships), but the factors/characteristics that could lead to bias. The authors are referring instead to a risk for bias, rather than measurement of actual bias.

Reviewer #3: Major comments

1. Selection of studies: How were these 13 guidelines selected for the Nuckols paper? Why were two WHO guidelines not included (2011 WHO guidelines on persisting pain in children (now withdrawn) and 2012 non-cancer pain in adults)). Did you include guidelines that examined both cancer and non-cancer pain? You variably refer to “guidelines” and “clinical practice guidelines”: did you restrict your cohort of studies to clinical guidelines only? All included studies appear to have been produced in and centered on the US or Canada. Were these the inclusion for the review? The inclusion and exclusion criteria for included guidelines needs to be clearly articulated, along with limitations of such criteria.

2. How did the Nuckols’ review do on AMSTAR tool? Ie what was the quality of this review? At the very least, the use of the Nuckols’ paper is a significant limitation as you did not verify the completeness and quality of their review. This needs to be addressed in the limitations.

3. Line 98: The focus is on 13 guideline panels: why not focus rather on 13 guidelines? Is there a one-to-one mapping of panel to guideline? If so, then you could use guidelines as the focus (they appear to be the unit of analysis anyway). In addition, the funder is separate from the guideline panel and there is one funder per guideline so the guideline focus makes more sense.

4. There is insufficient clarity on a number of key points and definitions:

a. Line 121: how define "conflicted organization"?

b. Table 2: how define COI? significant COI?

c. Search line 129: How did you define ”page” for your constraints on the google search? Doesn’t “page” depend on font size among other factors? The search terms are limited to “conflict of interest”: what about “conflict$ of interest$” “declarations of interest$” “disclosures…”, etc?

5. Throughout the manuscript the concepts of “conflict of interest” and “declarations of interest” appear to be conflated and confused. Authors disclose secondary interests which may be ad odds with the primary interest of the study or guideline; as a separate step, an independent person must decide of those declared interests represent conflicts (between the primary interests or duty, and the secondary interest eg. Financial gain).

a. For example (and not limited to);

i. Line 128: Journal articles generally only list declarations of interest: they leave the reader to decide if the disclosed interests are (significant) conflicts.

ii. Line 203: “conflicts of interest declarations”. Same point as above.

iii. Line 194

b. The authors need to make this distinction clear throughout the manuscript and use the appropriate term or phrase, as indicated.

6. Is GPR intended to be scored?

7. The AGREE-II “score” is used throughout this manuscript, eg lines 159 167, Table 3, etc. Although the AGREE-II authors mention using their tool in this way, most evidence synthesis and guidelines experts would never use a score as it means that each item in the tool is equally weighted, which they clearly are not. For example, a guideline can score highly on AGREE-II if they did everything right except did not do a systematic review of the evidence - obviously a fatal flaw and therefore a very poor quality guideline. Rather, each item or domain should be presented separately. At the very least, this paper must acknowledge this severe and misleading approach of using scores as a significant limitation.

8. The descriptive results are very simplistic and additional questions could be addressed with the same dataset the authors used. For example,

a. Were there differences across guideline sponsors? Government versus physician speciality organizations versus health systems?

b. Is there any relationship between AGREE "score" and number of flags (notwithstanding the limitations of scoring AGREE)?

c. Admittedly the small sample size precludes statistical analyses, but at least some of these questions could be raised for future consideration.

9. Line 209: How is <10% expertise in addiction medicine a good indicator of panel stacking? Lack of diverse views on a panel is much more complex a concept than expertise in addiction medicine or not. And how did you assess each individual's expertise?

10. Table 3 – could you provide more useful information on each guideline? Funder? Number of panel members no with DOI/COI? Types of COI: Professional versus receipt of industry funding for research or personal stock portfolios?

11. The conclusions in the discussion section and in the abstract overstate the findings in the study.

a. Line 240 – “Pharma had a significant influence on .. guidelines.,”

b. Line 328 “pharma had a pervasive influence…”

c. You have not in any way demonstrated that pharma had an influence, only that they had an apparent presence (and potential for influence).

d. You have demonstrated exposure but not causal influence: this distinction is critical to valid scientific discourse.

12. Throughout the discussion section the authors mention issues with the accuracy of disclosures and the authors mention doing google searches in the methods section, however there is nothing in the results section about accuracy of the disclosures. This disconnect either needs rectified or all references to inaccuracies as a finding need to be deleted

Minor comments

1. Page 4/21, lines 79-92: these two paragraphs on the selection of the GPR tool for assessing risk of bias and the use of the Nuckols review are related to methods and would be better situated in the methods section.

6. PLOS authors have the option to publish the peer review history of their article (what does this mean?). If published, this will include your full peer review and any attached files.

Reviewer #1: Yes: Lisa Bero

Reviewer #2: No

Reviewer #3: No

---

## [Author Response · Author response to Decision Letter 0]

4 Oct 2019

Responses to reviewers

Thank you for reviewing our manuscript. Please see our responses and changes below. 

In addition to the comments from the three reviewers I have some additional points that need to be addressed:

1. Do you have any information that these guidelines actually influenced prescribing behaviour? If they were largely ignored then it shouldn’t matter whether they were biased. On a related topic, if the guidelines influenced prescribing were they all equally influential? 

Response #1: In general guidelines have moderate impact on process measures. Impact on health outcomes is less clear. The evidence is low to moderate in quality. The new CDC guidelines are also associated with reduction in prescribing and unsafe prescribing. We have added this information to the introduction (p3, para 2): 

“They have a moderate impact on the behaviour of health care providers (1–3). A recent study found that the 2016 Centre for Disease Control (CDC) guideline for opioid prescribing led to declines in opioid prescribing (4).”

2. Both the GPR and the IOM recommendations came in after 2007. Were there recommendations about FCOI of panel members before that that the guidelines should have followed?

Response #2: There were a number of organizations that had recommendations for conflicts of interest. Most of these focused on disclosure. We have added this to the discussion (p15, para 2): 

“Although some of the guidelines were published prior to the 2011 recommendations from the Institute of Medicine, accurate disclosure was the accepted standard during the time period of these guidelines were published (5)”

3. How were the “red flag” definitions developed?

Response # 3 We operationalised the red flag definitions through consensus discussions. We have added included a table with the red flag definitions from Lenzer et al and our operationalized statements added to Table 2. We have also clarified in the manuscript (p7, para 4):

“We operationalized the GPR for our purposes by creating specific definitions for each red flag (Table 2). We accomplished this through several rounds of testing and discussion. Differences were resolved through discussion.”

Reviewer #1: This paper addresses a gap in the literature examining pharmaceutical industry involvement in the opioid epidemic – industry influence on clinical practice guidelines. It builds on a previous study examining opioid prescribing guidelines (for chronic non-cancer pain). As this earlier study had a number of weaknesses and the current study methods are vague, I think the paper needs substantial clarification before publication.

Introduction

Good description of the rationale for this paper. However, as this paper relies on the Nuckols review, I would like to see a more detailed critique of the pros and cons of this review. Why was it selected as the starting point, rather than the authors doing their own search and assessment of guidelines? Furthermore, this limits the study to only North American guidelines. this needs to be justified. 

Response #4 We selected the Nuckols review because it is the only systematic review and critical appraisal conducted on guidelines for the time period we were interested in analyzing—peak opioid prescribing. We have not found any conducted since. Although the review’s inclusion criteria included guidelines outside of North America, the English language restriction may have led to the exclusion of some guidelines that may have met the inclusion criteria. We have added a critique of the paper to the methods section. We have added this to the limitations section. 

Methods – Data Sources, P6, para 1:

“We included the same 13 guidelines for opioid prescribing for CNCP that were included in the Nuckols et al. 2014’s systematic review and critical appraisal (6). Nuckol’s et al. included English language guidelines published between 2007 and 2013 that addressed the use of opioids to treat CNCP in adults. The guidelines had to be clinical practice guidelines that included “recommendations intended to optimize patient care that are informed by a systematic review of evidence and an assessment of the benefits and harms of alternative care options” that were published after 2006 (7). To assess the quality of the Nuckols review, two authors (SMS and PL) independently rated the guidelines using the JBI Critical Appraisal Checklist for Systematic Reviews and Research Syntheses. Differences were resolved with discussion. The Nuckols review satisfied all checklist criteria except for item 7 (Were there methods to minimize errors in data extraction?). For item 7 the authors do not mention any methods to minimize errors in data extraction (e.g. data extraction in duplication or extraction double-checked). There were no concerns about the inclusion criteria, search strategy, criteria for appraising studies, appraisal method and methods to combine the data.” 

Limitations, p16, para 1:

“Our decision to build on the systematic review and appraisal by Nuckols et al., allowed us to build on past work. However, it also means we did not create and conduct our own search strategy. In particular, the English language requirement may have led to exclusion of some guidelines that would have otherwise met the inclusion criteria. “

Methods

It is difficult to evaluate the search strategy for guidelines as it is based on the Nuckols paper. I looked at the Nuckols paper and it is surprising they identified so few guidelines to include given their search strategy. 

Response #5. A description of the search strategy is included in the supplementary material linked from the study. Please also see response #4. 

There needs to be much more description of the GPR as this would help explain some of the decisions made about how nondisclosed financial ties were identified. 

Response #6 We have added more information to the methods section (p7, para 3):. 

“To guide our assessment of the risk of bias in guidelines, we used the Guideline Panel Review (GPR) (8). The GPR was created by an international expert working group (the Guideline Panel Review working group) in 2013 using a modified Delphi process (8). The tool assigns red flags to practice elements “known to introduce potential bias” (8). The goal of the group was to patients and inform those that develop, publish and use guidelines (Box 1). The GPR reflects the standards created by the Institute of Medicine for developing trustworthy guidelines (9). The GPR focuses on financial conflicts of interest, in particular those involving industry, because industry-funded networks dramatically amplify findings that support its products and obscure those that do not (8,10).” 

For example, page 6, line 124, it is not clear why searches for undisclosed conflicts of interests were made 2 years prior for the committee chairs and only in the current year for the committee members (line 127). Many studies look for undisclosed COI for 3 years prior to the index guideline (see Moynihan re undisclosed ties in NHMRC guidelines). The results (page 9) mention only the 2 year period.

Response #7 We opted for this time period because two years was a common reporting window for this time period

Institute of Medicine (US) Committee on Conflict of Interest in Medical Research, Education, and Practice; Lo B, Field MJ, editors.

Washington (DC): National Academies Press (US); 2009.

https://www.ncbi.nlm.nih.gov/books/NBK22943/

Table 2 shows how the authors operationalized the “red flags” in the GPR, but these are not the same as criteria used for the GPR. There should be a description of how the categories used in this paper were derived. 

Response # 8. We have added an explanation as to how we operationalized the GPR in the methods and more detail in a table. See also response #3. 

Page 7, line 157 mentioned a GPR score, but no scores are calculated or reported; the number of red flags is reported. 

Response # 9 We have changed this to state number of red flags 

Page 7, line 161. The authors should state why they extracted only the AGREEII scores and not the AMSTAR scores (presumably because AMSTAR applies to the systematic reviews used in the guidelines). Response #10 Please see methods

We did not extract the AMSTAR scores from the study because AMSTAR only assesses systematic reviews underpinning the guidelines, not the guideline panel. 

In addition, I am not sure that it is allowable for the authors to reprint the AMSTAR scores from Nuckols in this paper without permission. Nuckols is not published in an open access journal. 

Response #11 We are in the process of seeking permission. 

Results

My concerns with the results are primarily related to the use of the GPR.

Table 2 needs clarification. Table 2 should include the N in the title, which I assume is 13 guidelines? So, if I am interpreting the table correctly, 5 of 13 guidelines had a “red flag” for sponsor, meaning that the sponsor was a professional society that receives pharmaceutical industry funding or is a proprietary company. 

Response #12: We have added this clarification to the title and to the table (now table 3).

I disagree with the author’s definition of “Panel stacking” so think this needs further justification. I would argue that having panel members from multiple disciplines would promote diversity of views and minimize bias. However, the author rating suggests that there should be more members with expertise in addictions. 

Response #13 We have added an explanation why we used this approach in the methods section (p8, para 1) 

“The most challenging red flag to operationalize was panel or committee stacking because it was very difficult to determine every committee member’s views prior to the guideline development. This would be relatively simple to accomplish if the tool was used by the guideline organizer to self-report. However, it was not feasible for our assessment. Therefore, in discussion, we decided we would use expertise in chronic pain or addictions as a proxy for a balanced committee. This would ensure there were committee members who saw the benefits of opioid prescribing for CNCP and members who saw the harms. We set a minimum of 10% of committee members from each of these two categories..”

And the limitations section (p16, para 1):

“Another limitation was our difficulty in assessing for panel stacking. The GPR was designed with the idea that the guidelines committee would complete and submit the GPR along with the guidelines for publication. In this situation, the committee could easily and accurately assess the important views of the committee members prior to the creation of the guidelines. Conducting the assessment as external reviewers is much more difficult. However, we believe our approach (ensuring an adequate number of chronic pain and addictions specialists) would ensure that there were people on both sides of the opioid prescribing debate.”

The authors should define a major vs a minor role for a methodologist, especially as this issue is raised again in comparison to the AGREEII tool (page 12, lines 290-294). 

Response # 14 We have added this information to Table 2:

“No methodologist or methodologist has a minor role (i.e., not the chair or lead of the committee or lack of an oversight methodology committee)”

Conclusion

The comparison with the AGREEII tool, which highlights the AGREE limitations, is important and should be consolidated in one place in the discussion.

Response #15 We placed this comparison in para 3 of the discussion (p14). Please let us know if there is something else that should go in this paragraph as well:

“Our finding that guidelines that scored high on the 2014 critical appraisal by Nuckols et al. (6) had many red flags aligns with findings in a study by Eady et al. (11). In an appraisal of guidelines on treating acne, the authors reported that using the AGREE II tool during guideline development did not have “as great an effect on guideline quality as expected. There is considerable room for improvement in acne treatment guidelines in order to satisfy the Institute of Medicine’s trustworthiness criteria and avoid bias.” The AGREE II tool assesses conflicts of interests and strategies to mitigate bias differently than the GPR and the Institute of Medicine. The GPR assigns a red flag if the sponsoring organization or individuals on the guideline committees have a financial conflict of interest. The AGREE II, however, rates a guideline with conflicts of interest equally as one without, as long as they are reported and their potential impact on the guideline described (12). The medical literature indicates, however, that transparency is not sufficient to reduce bias and divestment from financial conflict is a better strategy (9,13–16). Additionally, Nuckols et al. did not use supplementary sources for conflicts of interests and therefore did not identify the four sponsoring organizations and three chairs who did not accurately report financial conflicts of interest with the pharmaceutical industry. This may have affected the AGREE II score. Finally, the AGREE II tool gives a guideline with a methodologist in a minor role a high score, whereas the GPR requires that the methodologist have a major role. The literature supports having a methodologist in a major role; when content experts take the lead, evidence reviews are less accurate (17–19). These findings indicate that the AGREE II may not accurately assess risk of bias from financial conflicts of interest with industry, particularly if appraisers do not seek out supplementary sources to assess for hidden conflicts of interest.” 

I would also highlight the extent of nondisclosure in the guidelines reviewed.

Response #16 We have highlighted this in para 4 of the discussion (p15): 

“Our finding that many guidelines either did not report or accurately report funding sources for the sponsoring organization and conflicts of interest for individuals is consistent with other studies and media reports (20–25). A recent ProPublica and New York Times investigation found that a top cancer researcher consistently failed to disclose millions in payments (26). Although some of the guidelines were published prior to the 2011 recommendations from the Institute of Medicine, accurate disclosure was the accepted standard during the time period of these guidelines were published (5). Reasons that individuals and organizations do not declare conflicts of interest has been poorly studied, but may include: not understanding what to declare (27,28); believing that a conflict is irrelevant (29); and concerns about reducing trust in the guideline user (29). Independent verification of conflicts of interest would help address reporting inaccuracies. This is easily done in the US with the Open Payments database (available since 2014) that records all industry payments to physicians (25). However, a similar database does not exist in Canada (30).”

Reviewer #2: I have provided comments in the attachment. My only comment here would be pertaining to #1 - that the authors should be cautious in their wording about bias and influence - the paper does not measure bias or influence (i.e., the real effects of the FCOI relationships), but the factors/characteristics that could lead to bias. The authors are referring instead to a risk for bias, rather than measurement of actual bias.

Response # 17 We have made this change throughout. 

Thank you for the opportunity to review your manuscript. This manuscript provides an important contribution to the literature, especially as we grow to understand the depths of industry involvement in the current opioid prescribing environment and resultant widespread risks for and real harm. Please find below my line-by-line comments. As a general comment, be sure to refer to risk for bias, rather than real bias because real bias was not measured. Accept with minor revisions.

- AS

Line No./Section Comment 

Introduction The population of guidelines is not explicitly stated. The paper refers to both Canadian and US guidelines at line 56, but a direct statement to this effect would help to clarify the scope of the study for the reader.

Response #18 The Nuckols’ guidelines inclusion criteria were English language guidelines for the use of opioids for the treatment of chronic non-cancer pain in adults. We have sought to make this clear throughout. (e.g., methods, p6, para 1) 

74, 90, 103, 113, 117, 118, 160, 168, 276, 288, 316 Nuckols et al vs. Nuckols et al. throughout – make sure that each reference is formatted consistently with the others.

Response #19 We have changed to Nuckols et al. throughout

92 Nuckol et al.’s – formatting of spacing and apostrophe to be revised

Response #20 We have reformatted. 

88, 98 In line 113, the authors state that they used the same 13 guidelines as in the Nuckols paper – is there a rationale for why additional new/updated guidelines were not included in this study beyond 2013 (i.e., the guidelines included in the Nuckols 2014 paper)? How was this decided and why?

Response #21

We decided to build on work already done. We were interested in the peak time of opioid prescribing. We wanted to be able to compare AGREEII scores conducted by another group to our assessment of bias. The decision was made as a group. 

88-90 These lines are written to sound like a coincidence that the chosen guidelines were also those that Nuckols analyzed, whereas it is the case that the Nuckols paper was the method of collection of the sample population to the exclusion of all others. The authors’ decision to study the same guidelines as in Nuckols and Nuckols as a tool of inclusion should be clearly stated and then justified (as in lines 90-91.

#22 Thank you. We have re-worded that section (Results, p10, para 1):

“We appraised the 13 guidelines in the systematic review and critical appraisal by Nuckols et al. (Table 1).”

174 What is the reason for including only one Canadian guideline? If the rationale is that the authors used Nuckols to identify the sample population, then it must be stated as such; otherwise, it reads repeatedly as though the authors chose and imposed their own exclusionary criteria on the guidelines to come to the same sample as Nuckols provided. (i.e., “The authors identified the population of guidelines by consulting Nuckols’ sample of guidelines. Nuckols included one Canadian and 12 American guidelines, so these were included in our analysis…”

#23 Thank you. We have sought to correct this throughout the document. (see response #22)

174 “…appraisal [no comma] (Table 1).”

#24Corrected

193 “…these five” – does this refer to those guidelines referred to in line 191? Perhaps reorganize paragraph’s sentences to clarify.

#25 Reworded

209 The authors don’t refer specifically to the content of guidelines, with the exception of addictions. It would be interesting and indeed useful to see the treatment categories of the 13 guidelines. Were the six guidelines addictions guidelines? Either way, clarify why addictions is stated here and for no other analysis of guidelines? Furthermore, what is the significance of members having expertise in addiction? Would this significance of members’ expertise apply respectively to the other guidelines?

#26 We have added information as to why we selected addictions as an important group to have on guideline committee. Please see response #13

224 Does this sentence intend that the inclusion of patient representatives somehow neutralize the conflicts or was it found that there were no COI (i.e., that the patient representatives did not have financial ties to industry) or that their COI were less important than others’, so could be excluded? Did the guidelines state that the patient representatives were independent from industry? Did the guideline identify their patient groups?

# 27 From the GPR publication, lack of patients increased risk of bias in guideline. Only some guidelines reported on the conflicts of interest of committee members, including patients. 

231 Rephrase, i.e., “This study likely provides a conservative quantification of potential for bias…”

#28 Added this statement (p16)

249 “Significant”: This study does not include statistical analysis, so perhaps remove the word “significant” and replace with more accurate word. 

#29 We have removed this word

249 “Influence”: This study did not quantify the bias, just the elements/characteristics most likely to increase the risk for bias - clarify 

#30 We have removed this word 

257 “some” – quantify

#31 have quantified this in results section

260 Add in: “…risk for pervasive influence…”

#32 Added 

277 Replace “another study” with “Eady et al. (60),…”

#33 Corrected

291 Spacing: AGREEII

# 34 Corrected 

291 Add: “The AGREE II tool gives…”

#35 Added 

291 Remove “And”

#36 Corrected

292 Add: “in a minor role a full (word choice, rephrase “full”)…

#37 Added

292 Change from “marks” to “score”

#38 Changed 

293 Add “supports having a methodologist…”

#39 Added

293 Change: “…major role, since…”

#40 changed

299 Individuals (remove comma)

#41 Removed

307 Remove “however”, begin sentence with “A…”

#42 changed

331 Word choice

#43 please assess

336 Change dashes to commas

#44 changed

Reviewer #3: Major comments

1. Selection of studies: How were these 13 guidelines selected for the Nuckols paper? Why were two WHO guidelines not included (2011 WHO guidelines on persisting pain in children (now withdrawn) and 2012 non-cancer pain in adults)). Did you include guidelines that examined both cancer and non-cancer pain? All included studies appear to have been produced in and centered on the US or Canada. Were these the inclusion for the review? The inclusion and exclusion criteria for included guidelines needs to be clearly articulated, along with limitations of such criteria. 

Response # 45 Please also see response # 4

The inclusion criteria only included adults. Additionally, the WHO 2011 guidelines on CNCP in adults in not a clinical practice guideline. It is directed at national policy-makers. Additionally, there were significant concerns that both guidelines were a marketing device with heavy industry influence. https://www.bmj.com/content/365/bmj.l2343.full

You variably refer to “guidelines” and “clinical practice guidelines”: did you restrict your cohort of studies to clinical guidelines only? 

Response #46 Nuckols only included clinical practice guidelines. We have added this to the text. We use the term guidelines to refer to these clinical practice guidelines to streamline text and avoid acronyms:

“Nuckol’s et al. included English language guidelines published between 2007 and 2013 that addressed the use of opioids to treat CNCP in adults. The guidelines had to be clinical practice guidelines that included “recommendations intended to optimize patient care that are informed by a systematic review of evidence and an assessment of the benefits and harms of alternative care options” that were published after 2006 (7).”

2. How did the Nuckols’ review do on AMSTAR tool? Ie what was the quality of this review? At the very least, the use of the Nuckols’ paper is a significant limitation as you did not verify the completeness and quality of their review. This needs to be addressed in the limitations.

Response #47 Please see response #4 

3. Line 98: The focus is on 13 guideline panels: why not focus rather on 13 guidelines? Is there a one-to-one mapping of panel to guideline? If so, then you could use guidelines as the focus (they appear to be the unit of analysis anyway). In addition, the funder is separate from the guideline panel and there is one funder per guideline so the guideline focus makes more sense. 

Response # 48 Thank you for this point. We have made this change. 

4. There is insufficient clarity on a number of key points and definitions:

a. Line 121: how define "conflicted organization"?

Response # 49 We have removed this term and used the term as defined in the table 

Sponsor is a professional society that receives pharmaceutical industry funding or is a proprietary company

b. Table 2: how define COI? significant COI? 

Response #50 

Response # 50 We added this information to the Table 2

c. Search line 129: How did you define ”page” for your constraints on the google search? Doesn’t “page” depend on font size among other factors? The search terms are limited to “conflict of interest”: what about “conflict$ of interest$” “declarations of interest$” “disclosures…”, etc? 

Response #51: Google displays 10 results per page as default. We have added this information to the manuscript. We did not search other terms. We have added to our limitations that is it possible that authors had conflicts of interest we did not find with our search. 

(p6): “For the Google search, we searched the author’s name with the term “conflict of interest.” We reviewed the first 30 entries (three pages) for information on conflicts at time of guideline publication and up to two years prior.”

(p16): “Another limitation is our Google search for conflicts of interest not reported in the guidelines. It is possible that we missed conflicts of interest by not conducting a more exhaustive search.”

5. Throughout the manuscript the concepts of “conflict of interest” and “declarations of interest” appear to be conflated and confused. Authors disclose secondary interests which may be ad odds with the primary interest of the study or guideline; as a separate step, an independent person must decide of those declared interests represent conflicts (between the primary interests or duty, and the secondary interest eg. Financial gain).

a. For example (and not limited to);

i. Line 128: Journal articles generally only list declarations of interest: they leave the reader to decide if the disclosed interests are (significant) conflicts.

ii. Line 203: “conflicts of interest declarations”. Same point as above.

iii. Line 194

b. The authors need to make this distinction clear throughout the manuscript and use the appropriate term or phrase, as indicated.

Response #52 Conflict of interest terminology does vary between organizations and journals. We opted to use the ICMJE terminology that asks all participants to self-report conflicts of interest. The organization places the responsibility on the authors to disclose actual, potential or perceived conflicts of interest. The editors make the management decision for the conflicts of interest. In our manuscript, we are not making management decisions but instead just the self-disclosed conflicts of interest (or in some cases the potential/actual/perceived conflicts of interest we found via a internet search). 

http://www.icmje.org/recommendations/browse/roles-and-responsibilities/author-responsibilities--conflicts-of-interest.html

6. Is GPR intended to be scored? 

Response #53 The authors use this terminology (i.e., “score”) in the article describing the GPR. We have amended the manuscript to remove this terminology and instead provide an appraisal or a count of red flags. 

7. The AGREE-II “score” is used throughout this manuscript, eg lines 159 167, Table 3, etc. Although the AGREE-II authors mention using their tool in this way, most evidence synthesis and guidelines experts would never use a score as it means that each item in the tool is equally weighted, which they clearly are not. For example, a guideline can score highly on AGREE-II if they did everything right except did not do a systematic review of the evidence - obviously a fatal flaw and therefore a very poor quality guideline. Rather, each item or domain should be presented separately. At the very least, this paper must acknowledge this severe and misleading approach of using scores as a significant limitation. 

Response #54 Nuckols et al. did not report on an average score but a global quality score (overall guideline assessment page 10 and page 40 AGREE II tool). It is one of the two final sections on the AGREE II tool meant to take into account the scores for each domain but not be an average score. This is the score we extracted. We have added this information to the manuscript. Thank you for noting this. 

(p9): “We also extracted the scores for AGREE II from Nuckols et al.’s systematic review and critical appraisal (6). We extracted the score for overall quality rating. This score is not an average score. It is a global score reported by reviewers after rating all the domains of quality. It is based on this information but is not calculated from those scores (6,12).” 

8. The descriptive results are very simplistic and additional questions could be addressed with the same dataset the authors used. For example,

a. Were there differences across guideline sponsors? Government versus physician speciality organizations versus health systems? 

Response #55 We did not extract this information as our focus was on financial conflicts of interest with the pharmaceutical industry. The Nuckols review does contain a broader description of the organizations. 

b. Is there any relationship between AGREE "score" and number of flags (notwithstanding the limitations of scoring AGREE)? c. Admittedly the small sample size precludes statistical analyses, but at least some of these questions could be raised for future consideration.

Response #56 Given the small numbers we did not assess for a relationship. However, we agree that work needs to be done to further assess how well the AGREEII tool and the GPR assess for risk of bias and how well they predict guideline content. 

9. Line 209: How is <10% expertise in addiction medicine a good indicator of panel stacking? Lack of diverse views on a panel is much more complex a concept than expertise in addiction medicine or not. And how did you assess each individual's expertise? 

Response #57 Please see response # 13

10. Table 3 – could you provide more useful information on each guideline? Funder? Number of panel members no with DOI/COI? Types of COI: Professional versus receipt of industry funding for research or personal stock portfolios? 

Response #58 We have added the appendix to this submission which contains some of this information. However, a detailed extraction was not in scope, rather our focus was applying the GPR tool, and the wording of the questions did not require full extraction. e.g., multiple committee members – once we found two, we did not need to specify for the others 

11. The conclusions in the discussion section and in the abstract overstate the findings in the study.

a. Line 240 – “Pharma had a significant influence on .. guidelines.,”

b. Line 328 “pharma had a pervasive influence…”

c. You have not in any way demonstrated that pharma had an influence, only that they had an apparent presence (and potential for influence).

d. You have demonstrated exposure but not causal influence: this distinction is critical to valid scientific d

Response # 59 Thank you for highlighting this. We have modified the manuscript to remove the word influence. (e.g., pervasive presence – p13).

11. Throughout the discussion section the authors mention issues with the accuracy of disclosures and the authors mention doing google searches in the methods section, however there is nothing in the results section about accuracy of the disclosures. This disconnect either needs rectified or all references to inaccuracies as a finding need to be deleted. 

Response # 60 Please see results section for this information (included in previous version) and appendix for the sources 

(P11, Sponsor):

“Among the four professional societies that accepted pharmaceutical industry funding, this information was not declared in the guidelines, but was found in supplementary sources.” 

(p11, Chair):

“In three of the five guidelines that received a red flag, conflicts of interest information was either not declared, or not stated, but we found financial conflicts of interest with the pharmaceutical industry in other source documents (31–33). “

Minor comments

1. Page 4/21, lines 79-92: these two paragraphs on the selection of the GPR tool for assessing risk of bias and the use of the Nuckols review are related to methods and would be better situated in the methods section. 

Response #61 We have moved most of this section to the methods. 

1. Lugtenberg M, Burgers JS, Westert GP. Effects of evidence-based clinical practice guidelines on quality of care: a systematic review. Qual Saf Health Care. 2009 Oct;18(5):385–92. 

2. Grimshaw JM, Russell IT. Effect of clinical guidelines on medical practice: a systematic review of rigorous evaluations. The Lancet. 1993 Nov 27;342(8883):1317–22. 

3. Spooner L, Fernandes K, Martins D, Juurlink D, Mamdani M, Paterson JM, et al. High-Dose Opioid Prescribing and Opioid-Related Hospitalization: A Population-Based Study. PLOS ONE. 2016 Dec 14;11(12):e0167479. 

4. Bohnert ASB, Guy GP, Losby JL. Opioid Prescribing in the United States Before and After the Centers for Disease Control and Prevention’s 2016 Opioid Guideline. Ann Intern Med. 2018 Sep 18;169(6):367. 

5. Boyd EA, Bero LA. Improving the use of research evidence in guideline development: 4. Managing conflicts of interests. Health Res Policy Syst. 2006 Dec 1;4:16. 

6. Nuckols TK, Anderson L, Popescu I, Diamant AL, Doyle B, Di Capua P, et al. Opioid prescribing: a systematic review and critical appraisal of guidelines for chronic pain. Ann Intern Med. 2014 Jan 7;160(1):38–47. 

7. Shekelle P, Woolf S, Grimshaw JM, Schünemann HJ, Eccles MP. Developing clinical practice guidelines: reviewing, reporting, and publishing guidelines; updating guidelines; and the emerging issues of enhancing guideline implementability and accounting for comorbid conditions in guideline development. Implementation Science. 2012 Jul 4;7(1):62. 

8. Lenzer J, Hoffman JR, Furberg CD, Ioannidis JPA. Ensuring the integrity of clinical practice guidelines: a tool for protecting patients. BMJ. 2013 Sep 17;347:f5535. 

9. Institute of Medicine (US) Committee on Standards for Developing Trustworthy Clinical Practice Guidelines. Clinical Practice Guidelines We Can Trust [Internet]. Graham R, Mancher M, Miller Wolman D, Greenfield S, Steinberg E, editors. Washington (DC): National Academies Press (US); 2011 [cited 2019 Mar 26]. Available from: http://www.ncbi.nlm.nih.gov/books/NBK209539/

10. Bero LA, Grundy Q. Why Having a (Nonfinancial) Interest Is Not a Conflict of Interest. PLOS Biology. 2016 Dec 21;14(12):e2001221. 

11. Eady EA, Layton AM, Sprakel J, Arents BWM, Fedorowicz Z, Zuuren EJ van. AGREE II assessments of recent acne treatment guidelines: how well do they reveal trustworthiness as defined by the U.S. Institute of Medicine criteria? British Journal of Dermatology. 2017;177(6):1716–25. 

12. Brouwers MC, Kho ME, Browman GP, Burgers JS, Cluzeau F, Feder G, et al. AGREE II: advancing guideline development, reporting and evaluation in health care. CMAJ. 2010 Dec 14;182(18):E839–42. 

13. Wilson M. Is transparency really a panacea? J R Soc Med. 2014 Jun 1;107(6):216–7. 

14. Wilson M. The Sunshine Act: Commercial conflicts of interest and the limits of transparency. Open Med. 2014 Jan 14;8(1):e10–3. 

15. Loewenstein G SS. The unintended consequences of conflict of interest disclosure. JAMA. 2012 Feb 15;307(7):669–70. 

16. Melo-Martín I de, Intemann K. How do disclosure policies fail? Let us count the ways. FASEB J. 2009 Jun 1;23(6):1638–42. 

17. Panagiotou OA, Ioannidis JPA. Primary study authors of significant studies are more likely to believe that a strong association exists in a heterogeneous meta-analysis compared with methodologists. J Clin Epidemiol. 2012 Jul;65(7):740–7. 

18. Oxman AD, Guyatt GH. The science of reviewing research. Ann N Y Acad Sci. 1993 Dec 31;703:125–33; discussion 133-134. 

19. Antman EM, Lau J, Kupelnick B, Mosteller F, Chalmers TC. A comparison of results of meta-analyses of randomized control trials and recommendations of clinical experts. Treatments for myocardial infarction. JAMA. 1992 Jul 8;268(2):240–8. 

20. Checketts JX, Sims MT, Vassar M. Evaluating Industry Payments Among Dermatology Clinical Practice Guidelines Authors. JAMA Dermatol. 2017 01;153(12):1229–35. 

21. Bauchner H, Fontanarosa PB, Flanagin A. Conflicts of Interests, Authors, and Journals: New Challenges for a Persistent Problem. JAMA. 2018 Dec 11;320(22):2315–8. 

22. Boddapati V, Fu MC, Nwachukwu BU, Ranawat AS, Zhen WY, Dines JS. Accuracy Between AJSM Author-Reported Disclosures and the Centers for Medicare and Medicaid Services Open Payments Database. Am J Sports Med. 2018 Mar;46(4):969–76. 

23. Horn J, Checketts JX, Jawhar O, Vassar M. Evaluation of Industry Relationships Among Authors of Otolaryngology Clinical Practice Guidelines. JAMA Otolaryngol Head Neck Surg. 2018 Mar 1;144(3):194–201. 

24. Lopez J, Samaha G, Purvis TE, Siegel G, Jabbari J, Ahmed R, et al. The Accuracy of Conflict-of-Interest Disclosures Reported by Plastic Surgeons and Industry. Plast Reconstr Surg. 2018;141(6):1592–9. 

25. Ziai K, Pigazzi A, Smith BR, Nouri-Nikbakht R, Nepomuceno H, Carmichael JC, et al. Association of Compensation From the Surgical and Medical Device Industry to Physicians and Self-declared Conflict of Interest. JAMA Surg. 2018 Nov 1;153(11):997–1002. 

26. Ornstein C, Thomas K. Top Cancer Researcher Fails to Disclose Corporate Financial Ties in Major Research Journals. The New York Times [Internet]. 2019 Jan 12 [cited 2019 May 13]; Available from: https://www.nytimes.com/2018/09/08/health/jose-baselga-cancer-memorial-sloan-kettering.html

27. Rohwer A, Young T, Wager E, Garner P. Authorship, plagiarism and conflict of interest: views and practices from low/middle-income country health researchers. BMJ Open [Internet]. 2017 Nov 22 [cited 2019 May 14];7(11). Available from: https://www.ncbi.nlm.nih.gov/pmc/articles/PMC5719292/

28. Baethge C. The effect of a conflict of interest disclosure form using closed questions on the number of positive conflicts of interest declared – a controlled study. PeerJ [Internet]. 2013 Aug 13 [cited 2019 May 14];1. Available from: https://www.ncbi.nlm.nih.gov/pmc/articles/PMC3746959/

29. Weinfurt KP, Friedman JY, Dinan MA, Allsbrook JS, Hall MA, Dhillon JK, et al. Disclosing Conflicts of Interest in Clinical Research: Views of Institutional Review Boards, Conflict of Interest Committees, and Investigators. J Law Med Ethics. 2006;34(3):581–481. 

30. Owens B. Ontario delays implementation of pharma transparency rules. CMAJ. 2019 Feb 25;191(8):E241–2. 

31. Department of Veterans Affairs, Department of Defense. Clinical Practice Guideline for Management of Opioid Therapy for Chronic Pain [Internet]. Department of Defense; 2012 [cited 2016 Jun 27]. Available from: http://www.healthquality.va.gov/guidelines/Pain/cot/COT_312_Full-er.pdf

32. American Geriatrics Society Panel on Pharmacological Management of Persistent Pain in Older Persons. Pharmacological management of persistent pain in older persons. J Am Geriatr Soc. 2009 Aug;57(8):1331–46. 

33. Fine PG, Portenoy RK, Ad Hoc Expert Panel on Evidence Review and Guidelines for Opioid Rotation. Establishing “best practices” for opioid rotation: conclusions of an expert panel. J Pain Symptom Manage. 2009 Sep;38(3):418–25.

---

## [Decision Letter · Decision Letter 1]

16 Oct 2019

PONE-D-19-17372R1

Roots of the opioid crisis: an appraisal of financial conflicts of interest in clinical practice guideline panels at the peak of opioid prescribing

PLOS ONE

Dear Dr. Spithoff,

Thank you for submitting your manuscript to PLOS ONE. After careful consideration, we feel that it has merit but does not fully meet PLOS ONE’s publication criteria as it currently stands. Therefore, we invite you to submit a revised version of the manuscript that addresses the points raised during the review process.

In addition to the comments from Reviewer 2 I have some additional points. Reviewer 2 used the line numbers from the marked up revision rather than the clean copy and so to avoid confusion I’ll also use the line number in the marked up copy.

Line 62: Also include the generic name as well as the brand name.Lines 214-216: The comment by reviewer 2 about these lines appears to be due to problems reading the marked up text and do not have to be addressed.By using the systematic review by Nuckols the authors are accepting that this review uncovered all of the English language guidelines that existed at the time, but the authors should acknowledge that Nuckols may have missed English language guidelines.

We would appreciate receiving your revised manuscript by Nov 30 2019 11:59PM. To enhance the reproducibility of your results, we recommend that if applicable you deposit your laboratory protocols in protocols.io, where a protocol can be assigned its own identifier (DOI) such that it can be cited independently in the future. For instructions see: http://journals.plos.org/plosone/s/submission-guidelines#loc-laboratory-protocols

We look forward to receiving your revised manuscript.

Kind regards,

Joel Lexchin, MD

Academic Editor

PLOS ONE

Reviewers' comments:

Reviewer's Responses to Questions

**Comments to the Author**

1. If the authors have adequately addressed your comments raised in a previous round of review and you feel that this manuscript is now acceptable for publication, you may indicate that here to bypass the “Comments to the Author” section, enter your conflict of interest statement in the “Confidential to Editor” section, and submit your "Accept" recommendation.

Reviewer #2: All comments have been addressed

Reviewer #3: (No Response)

2. Is the manuscript technically sound, and do the data support the conclusions?

Reviewer #2: Yes

Reviewer #3: Partly

3. Has the statistical analysis been performed appropriately and rigorously? 

Reviewer #2: N/A

Reviewer #3: N/A

4. Have the authors made all data underlying the findings in their manuscript fully available?

Reviewer #2: Yes

Reviewer #3: Yes

5. Is the manuscript presented in an intelligible fashion and written in standard English?

Reviewer #2: Yes

Reviewer #3: Yes

6. Review Comments to the Author

Reviewer #2: (No Response)

Reviewer #3: Major comments

1. The authors have largely addressed my concerns. However, two important issues remain.

2. Lines 215-216: “Therefore, in discussion, we decided we would use expertise in have with chronic pain or addiction s as a proxy for a balanced committee expertise.” . The explanation provided still does not make sense to me.

3. Line 243-244: “We extracted the score for overall quality rating. This score is not an average score. It is a global score reported by reviewers ’after rating all the domains of quality. It is based on this information but is not calculated from those scores”. This verbiage remains unclear – it appears you are using the overall qualitative assessment suggested for AGREE-II, which is not a “score” as the latte implies a quantitative value, does it not?

Minor comments

1. Lines 97 to 108 appear to be primarily methods and not introduction. In addition they appear to be redundant with information on page 7. Consider revising and deleting some of the text in the introduction.

2. Lines 177-178 – some text appears to be missing.

7. PLOS authors have the option to publish the peer review history of their article (what does this mean?). If published, this will include your full peer review and any attached files.

Reviewer #2: No

Reviewer #3: No

---

## [Author Response · Author response to Decision Letter 1]

10 Dec 2019

Reponses to reviewers Nov 29 2019 

Dear reviewers,

Thank you for reviewing our draft manuscript. The suggested revisions have strengthened the paper. 

We were unable to get permission from Annals of Internal medicine to reproduce and publish the Nuckols data in our table under a CC BY licence. Therefore, we have removed the information from Table 4. and cited it in the text in the results section instead. The changes to the text are highlighted in yellow. 

In addition to the comments from Reviewer 2 I have some additional points. Reviewer 2 used the line numbers from the marked up revision rather than the clean copy and so to avoid confusion 

I’ll also use the line number in the marked up copy.

1. Line 62: Also include the generic name as well as the brand name.

We have modified this to include (oxycodone)

2. Lines 214-216: The comment by reviewer 2 about these lines appears to be due to problems reading the marked up text and do not have to be addressed.

3. By using the systematic review by Nuckols the authors are accepting that this review uncovered all of the English language guidelines that existed at the time, but the authors should acknowledge that Nuckols may have missed English language guidelines.

We have added this to the limitations section. 

“It is also possible that Nuckols et al. missed English language guidelines (despite an appropriate search strategy).”

Reviewer #3: Major comments

1. The authors have largely addressed my concerns. However, two important issues remain.

2. Lines 215-216: “Therefore, in discussion, we decided we would use expertise in have with chronic pain or addiction s as a proxy for a balanced committee expertise.” 

The explanation provided still does not make sense to me.

We have deferred to editor (see above) for this comment

3. Line 243-244: “We extracted the score for overall quality rating. This score is not an average score. It is a global score reported by reviewers ’after rating all the domains of quality. It is based on this information but is not calculated from those scores”. 

This verbiage remains unclear – it appears you are using the overall qualitative assessment suggested for AGREE-II, which is not a “score” as the latte implies a quantitative value, does it not?

The AGREE-II asks the reviewer to assign an overall quality rating as a number between 1 and 7. The user manual on page 8 refers to assigning a global rating and instructs reviewer on when to give a score of 1 or 7, or in between. 

We have added throughout the document that the scores are the scores for an overall quality rating. 

Minor comments

1. Lines 97 to 108 appear to be primarily methods and not introduction. In addition they appear to be redundant with information on page 7. Consider revising and deleting some of the text in the introduction.

We have removed these lines from the introduction and modified it. 

“Our study objective was to appraise the risk of bias from financial conflicts of interest with the pharmaceutical industry in the guidelines included in the 2014 systematic review and critical appraisal by Nuckols et al. (25).” 

2. Lines 177-178 – some text appears to be missing.

We have fixed the sentence. 

“The goal of the group was to inform patients and those that develop, publish and use guidelines.”

---

## [Editor Report · Decision Letter 2]

12 Dec 2019

Drivers of the opioid crisis: an appraisal of financial conflicts of interest in clinical practice guidelines at the peak of opioid prescribing

PONE-D-19-17372R2

Dear Dr. Spithoff,

We are pleased to inform you that your manuscript has been judged scientifically suitable for publication and will be formally accepted for publication once it complies with all outstanding technical requirements.

With kind regards,

Joel Lexchin, MD

Academic Editor

PLOS ONE
---

## [Editor Report · Acceptance letter]

15 Jan 2020

PONE-D-19-17372R2 

Drivers of the opioid crisis: an appraisal of financial conflicts of interest in clinical practice guideline panels at the peak of opioid prescribing 

Dear Dr. Spithoff:

I am pleased to inform you that your manuscript has been deemed suitable for publication in PLOS ONE. Congratulations! Your manuscript is now with our production department. 

With kind regards,

on behalf of

Prof. Joel Lexchin 

Academic Editor

PLOS ONE